# Three-dimensional spin-wave dynamics, localization and interference in a synthetic antiferromagnet

Davide Girardi [1], Simone Finizio [2], Claire Donnelly [3,4], Guglielmo Rubini[1], Sina Mayr[2,5], Valerio Levati [1], Simone Cuccurullo[1], Federico Maspero [1], Jörg Raabe [2], Daniela Petti [1] ✉ & Edoardo Albisetti [1] ✉

Spin waves are collective perturbations in the orientation of the magnetic moments in magnetically ordered materials. Their rich phenomenology is intrinsically three-dimensional; however, the three-dimensional imaging of spin waves has so far not been possible. Here, we image the three-dimensional dynamics of spin waves excited in a synthetic antiferromagnet, with nanoscale spatial resolution and sub-ns temporal resolution, using time-resolved magnetic laminography. In this way, we map the distribution of the spin-wave modes throughout the volume of the structure, revealing unexpected depth-dependent profiles originating from the interlayer dipolar interaction. We experimentally demonstrate the existence of complex three-dimensional interference patterns and analyze them via micromagnetic modelling. We find that these patterns are generated by the superposition of spin waves with non-uniform amplitude profiles, and that their features can be controlled by tuning the composition and structure of the magnetic system. Our results open unforeseen possibilities for the study and manipulation of complex spin-wave modes within nanostructures and magnonic devices.

The three-dimensional nature of wave phenomena in condensed matter is a key aspect in many disciplines across nanoscience, from acoustics[1] to nano-optics[2–4] to plasmonics[5], and for their applications in technology. In spin waves[6–8], three-dimensionality is inherently connected with their phenomenology. For example, the chiral nature of magnetostatic surface spin-wave modes[9] leads to a non-uniform amplitude profile through the thickness of the film and immunity from backscattering[10]. Moreover, higher-order propagating modes[11,12], perpendicular standing waves[13] and modes supported by complex spin textures[14,15] display a wealth of distinctive three-dimensional features. Beyond thin films, the morphology of three-dimensional magnetic nanostructures[16–18] and curvilinear systems[19,20] can induce novel effects in the spin-wave properties. Such aspects are crucial for the young field

of three-dimensional magnonics[21] which, following the trend of electronics and photonics, aims to harness the third dimension to realize novel functionalities in devices. In this framework, spin-wave interference has been widely exploited for Boolean and non-Boolean SW computing[22–25], and moving towards the third dimension holds promise for vertically integrated platforms.

In the past decades, optical and X-ray techniques such as Brillouin Light Scattering (BLS)[26,27], time-resolved Kerr Effect[28,29], and X-ray microscopy[30,31] allowed to spatially map the spin-wave properties in two dimensions, across the plane of the films. However, as these techniques are two-dimensional, three-dimensional imaging of spin waves has not been possible, so that experimental insight in their three-dimensional nature has not yet been achieved.

[1]Dipartimento di Fisica, Politecnico di Milano; Piazza Leonardo da Vinci 32, Milano 20133, Italy. [2]Swiss Light Source, Paul Scherrer Institut; Forschungsstrasse 111 5232 PSI, Villigen, Switzerland. [3]Max Planck Institute for Chemical Physics of Solids; Nöthnitzer Str. 40, 01187 Dresden, Germany. [4]International Institute for Sustainability with Knotted Chiral Meta Matter (WPI-SKCM2), Hiroshima University, Hiroshima 739-8526, Japan. [5]Laboratory for Mesoscopic Systems, Department of Materials, ETH Zurich, 8093 Zurich, Switzerland. ✉e-mail: daniela.petti@polimi.it; edoardo.albisetti@polimi.it

Here, we reveal the full three-dimensional structure of spin waves propagating in a synthetic antiferromagnet (SAF), by reconstructing the dynamics of the magnetization via Time-Resolved Soft X-ray Laminography (TR-SoXL)[32,33]. SAFs hold promise in spintronics and magnonics, due to the possibility to efficiently excite non-reciprocal short-wavelength spin waves using spin textures such as vortices or domain walls, which propagate for multiple wavelengths giving rise to robust interference figures[30,34,35]. Interestingly, in SAF systems, spin waves have been predicted >40 years ago to possess complex depth-dependent features[36], which are expected to have a major impact on their propagation and interference, and have not been directly observed so far.

By mapping both the in-plane and out-of-plane components of the magnetization, we fully reconstruct the precessional dynamics associated to propagating spin waves, with nanoscale spatial resolution across the plane and through the volume of the sample. We find non-uniform spin-wave mode profiles, indicating the localization of the spin waves, that lead to the generation of complex three-dimensional features due to spin-wave interference, which we reveal experimentally and analyze with micromagnetic simulations.

The investigated SAF system consists of a CoFeB 50 / Ru 0.5 / NiFe 40 / Ru 4 (nm) rectangular $2 \times 3 \mu m^2$ microstructure (Fig. 1c), fabricated on X-ray transparent SiN membranes. The two ferromagnetic CoFeB and NiFe layers are coupled antiferromagnetically thanks to the presence of a thin Ru interlayer, as shown in Fig. 1b. To excite spin waves, a Cu radiofrequency antenna was fabricated on top of the structure (Fig. 1a). The coupling between the RF Oersted magnetic field generated by the stripline and the spin textures in the microstructure allows for the efficient generation of spin waves. Additionally, in SAFs, the interlayer exchange and dipolar interaction gives rise to coupled nonreciprocal spin-wave modes in the two layers. In particular, at our experimental excitation frequency of 0.86 GHz, chosen to maximize the spin-wave excitation efficiency, we expect to have only acoustic spin waves, characterized by antiparallel in-plane / parallel out-of-plane magnetization dynamics[30,36]. Such nonreciprocal acoustic modes are sketched in Fig. 1d with $\mathbf{k_{sw}}$ indicating the propagation direction. The strong nonreciprocity of the investigated SAF is further analyzed by the simulated dispersion relation shown in Supplementary Fig. 3, and confirms the unidirectionality of the spin-wave propagation at our experimental excitation frequency. To quantitatively analyze the spin-wave properties, we define the in-plane and out-of-plane dynamic angles $\Delta\theta$ ($t$) and $\Delta\varphi$ ($t$) as the in-plane and out-of-plane tilt of the dynamic magnetization with respect to its static direction, indicated by the black arrow in Fig. 1e (see also Supplementary Fig. 4). In this framework, for small angles, spin waves appear as sinusoidal oscillations of $\Delta\theta$ and $\Delta\varphi$ in space and time.

To map spin waves in three dimensions in our SAF, we performed Time-Resolved Magnetic Laminography, by acquiring time-resolved images of the magnetization dynamics[32,33]. 37 different projections of such images were acquired by rotating the sample around the laminography axis, keeping the incident X-ray beam at a fixed angle with respect to the sample, as shown in Fig. 1a. This configuration allowed us to be sensitive to all three components of the magnetization. To probe the NiFe and CoFeB layer separately, we acquired two sets of projections by tuning the X-ray energy to the Ni and Co edges, respectively. The full dynamics of the magnetization vector was then reconstructed in three dimensions in each layer separately using an iterative algorithm[37]. Experimental details are reported in the methods section.

A snapshot of the three-dimensional reconstruction of the magnetization dynamics in the whole sample is shown in Fig. 1f, where the arrows indicate the magnetization direction across the volume, and the color code shows the in-plane dynamic angle $\Delta\theta$, associated to the spin waves propagating along $\mathbf{k_{sw}}$. We estimate the spin-wave amplitude as the maximum value of $\Delta\theta \sim 2°$, suggesting that the observed

SWs consist of small dynamic perturbations over the static magnetization configuration (see Supplementary Fig. 2). By analyzing the static magnetic configuration, we observe that due to the shape anisotropy, the magnetization lies predominantly parallel to the surface of the films and gradually curls clockwise (counterclockwise) in the NiFe (CoFeB) layer, maintaining the expected antiferromagnetic coupling, which is crucial for supporting the acoustic spin-wave modes.

Together with the curling of the magnetization, we observe the stabilization of nanoscale spin textures[38]. In particular, a vortex core is observed in the central region, and a sharp domain wall cuts across the rectangular structure from edge to edge. In this respect, we observe that the vortex dynamics and domain wall oscillation excited by the RF antenna are the main sources of spin waves[34,35], which propagate from the center towards the edges, with no sizeable back-reflection, as expected from the non-reciprocal spin-wave dispersion in SAFs[30,35] (see experimental Supplementary Movies 5, 6). Such a complex system, combining multiple spin-wave emitters and regions with quasi-uniform magnetization, allows us to investigate both free-space propagation and spin-wave interference.

By measuring the dynamics of all three components of the magnetization at the nanoscale, we directly image the precession of the magnetization vector associated to spin-wave modes, revealing experimentally for the first time their full anatomy. For doing so, we focus on a region of the sample with quasi-uniform magnetization (see Fig. 1f), where spin waves emitted by a domain wall propagate freely from right to left (Fig. 2b). Here, we observe the phase difference in the precession of neighboring moments along one wavelength in both NiFe (Fig. 2a) and CoFeB (Fig. 2c), which confirms the propagating character of the spin waves. To study in details the modes, we extract horizontal spatial profiles of the in-plane dynamics in both layers (Fig. 2d). We observe a sinusoidal oscillation with wavelength ~ 650 nm, in agreement with simulations (Supplementary Fig. 3), and point-by-point antiparallel orientation of the in-plane components in the two layers, a signature of the "acoustic" propagating spin waves.

To study the full temporal evolution of the propagating mode, we reconstruct time-resolved sequences (Supplementary Movies 1-4), where the magnetization orientation is shown every 0.17 ns, over one spin-wave period. Remarkably, we directly visualize the precessional trajectory and "right-hand" chirality of spin waves in the time domain, as shown by the ~ π/2 phase difference between the in-plane and out-of-plane time-traces of Fig. 2e. Such vectorial, time-resolved imaging could be used to gain a detailed insight on the magnetization precession in other systems, such as natural[39] and artificial[40] ferrimagnets or antiferromagnets, where the magnon chirality can represent an additional degree of freedom.

In order to study the mode profiles, we now take advantage of three-dimensional imaging to investigate the properties of spin waves through the thickness of each individual layer. For doing so, we focus on another region of the sample, shown on the right of Fig. 1f, where planar wavefronts are emitted by a domain wall, and propagate towards the edge of the structure, in a region with quasi-uniform magnetization (Fig. 3a and corresponding time-resolved Supplementary Movies 7, 8). To study the spin-wave properties through the thickness and along the propagation direction, we map the in-plane and out-of-plane spin-wave amplitudes $A_{\Delta\theta}$ and $A_{\Delta\varphi}$, in correspondence of the green vertical cross-section of the sample (see Methods). In particular, Fig. 3b,c report the data points corresponding to the green cross-section of Fig. 3a, visualized with a 20 nm x 20 nm pixel size in the horizontal and vertical directions. Interestingly, we observe strongly non-uniform profiles through the thickness, where in-plane (Fig. 3b, d) and out-of-plane (Fig. 3c, e) amplitudes display a significantly different depth-dependence. In particular, we observe that the in-plane amplitude is maximum at the top NiFe surface, and decreases monotonically through the structure, reaching its minimum

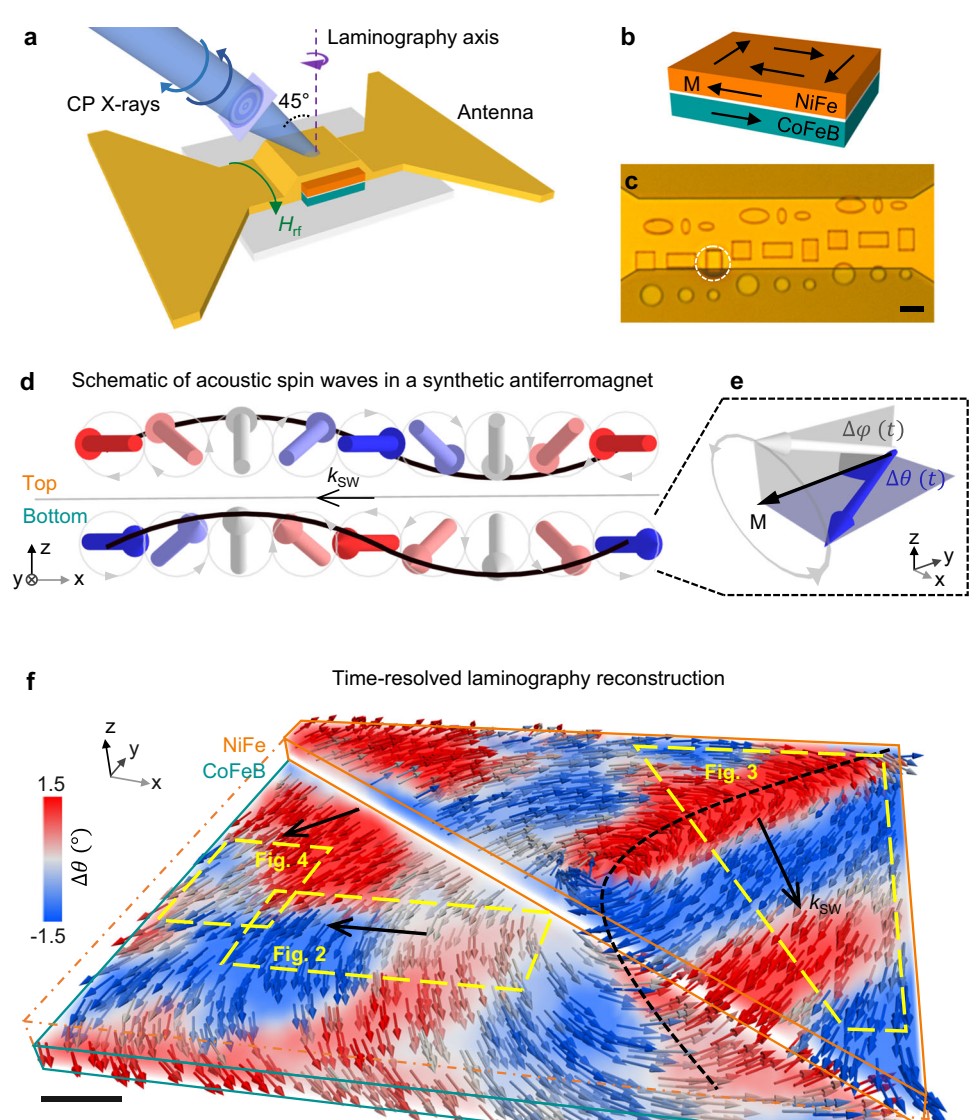

**Fig. 1 | Experimental setup and time-resolved three-dimensional reconstruction of spin waves in a synthetic antiferromagnet. a** Sketch of the laminography setup. The sample is tilted at 45° with respect to the incident circular polarized (CP) X-rays. Time-resolved projections with magnetic contrast are acquired at several rotation angles around the laminography axis (in purple). An oscillating magnetic field $\mathbf{H_{rf}}$ generated by the antenna excites spin waves in the synthetic antiferromagnet. **b** The magnetization (black arrows) in the CoFeB 50 / Ru 0.5 / NiFe 40 (nm) microstructure is antiparallel in the two ferromagnetic layers, and curls in a flux-closure configuration. **c** The white circle highlights the investigated microstructure in the optical image. Scale bar, 3 μm. **d** Schematic of nonreciprocal acoustic spin waves propagating in a synthetic antiferromagnet. The magnetization (arrows) precesses with a spatial phase difference along the propagation direction $k_{SW}$. The in-plane dynamic magnetization (blue-red coloring), is always antiparallel in the two layers, while the out-of-plane dynamic magnetization is parallel. **e** The dynamic angles $\Delta\theta\,(t)$ and $\Delta\varphi\,(t)$ are defined as the time-dependent in-plane ($\Delta\theta$) and out-of-plane ($\Delta\varphi$) tilt of the magnetization with respect to the static direction, respectively. **f** Experimental reconstruction of the three-dimensional magnetization dynamics (arrows) excited at 0.86 GHz in NiFe and CoFeB. In this snapshot, spin waves are visualized as sinusoidal spatial oscillations of the in-plane magnetization dynamics ($\Delta\theta$, blue-red coloring). The regions of the sample studied in each figure are indicated in yellow. On the right, planar spin waves emitted by a domain wall (dashed line) propagate along $\mathbf{k_{SW}}$. On the left, spin waves propagating at an angle interfere in the dashed rectangular region corresponding to Fig. 4. Scale bar, 200 nm.

at the bottom CoFeB surface. Contrarily, the out-of-plane SW amplitude is maximum in the middle of the structure and decreases towards both the top NiFe and bottom CoFeB surfaces. This leads to two effects on the geometry of the modes. First, a qualitatively different localization of the in-plane and out-of-plane spin-wave dynamics is observed at the top surface and middle of the structure, respectively. Second, a complex deformation of the precession through the thickness, which is high-amplitude and dominated by the in-plane component at the top surface and gradually becomes more circular in the middle due to the reduction of the in-plane dynamics and increase of the out-of-plane dynamics. Finally, it is low-

amplitude at the bottom surface, where both in-plane and out-of-plane dynamics are minimum. The results are nicely reproduced by simulations in the insets of (Fig. 3d, e).

The origin of such peculiar three-dimensional profiles can be ascribed to the asymmetry in the dipolar dynamic fields generated by the different magnetic moments of the NiFe and CoFeB layers. In fact, since the magnetic moment of NiFe is lower than CoFeB, a higher spin-wave amplitude is required for guaranteeing flux closure. At the same time, the presence of the interlayer exchange coupling in our SAF system tends to favor a matching of the amplitudes at the interface, resulting in a monotonic decrease from the top NiFe to the bottom

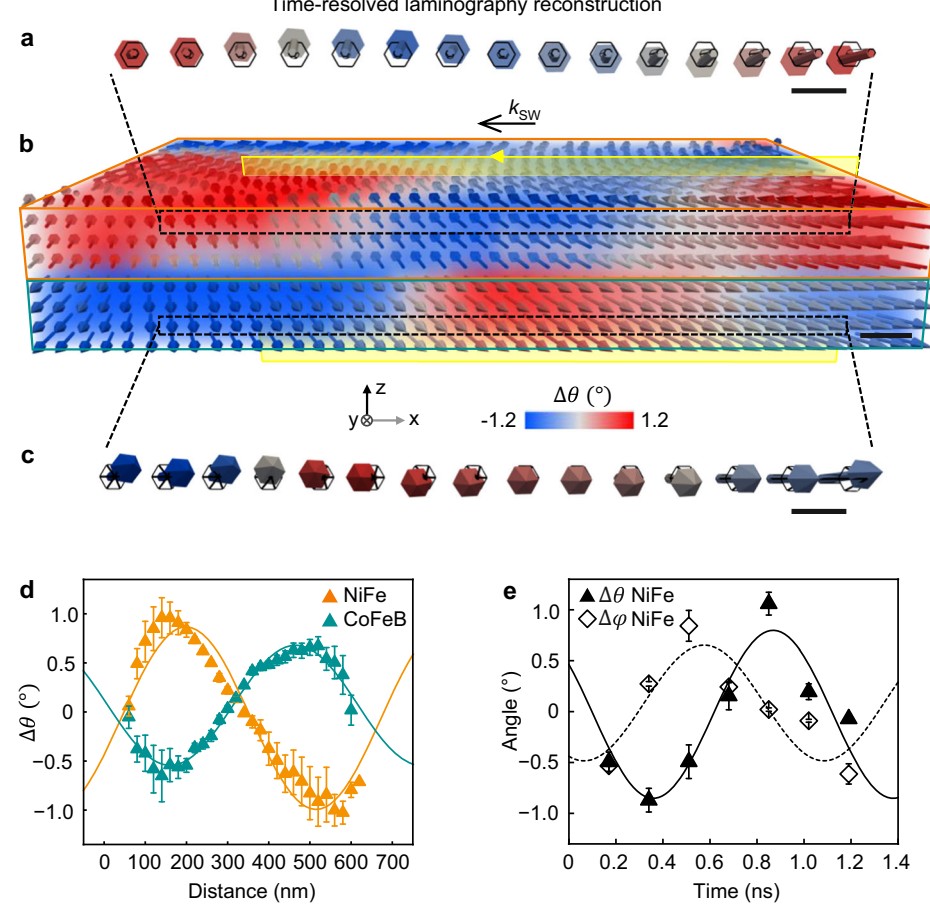

**Fig. 2 | Three-dimensional dynamics of the acoustic spin-wave mode. a–c** In this region of the sample (**b**) coherent spin waves propagate freely along $k_{SW}$, akin to the schematic of Fig. 1d. By mapping the dynamics of all three components of the magnetization in NiFe (**a**) and CoFeB (**c**) the "right-hand" precessional motion (full arrows) around the static magnetization direction (black arrows), and the periodic spatial phase difference associated to coherent wave propagation, are directly observed. (See corresponding time-resolved Supplementary Movies 2, 4 for the full dynamics). **d** The antiparallel coupling of the in-plane dynamics $\Delta\theta$, signature of the acoustic modes, is highlighted by the averaged profiles extracted along the yellow plane in (**b**) for the whole thickness of each layer, from NiFe (orange) and CoFeB (blue), showing sinusoidal oscillations in antiphase, and a spin-wave wavelength of ~650 nm. **e** Time-traces of the in-plane $\Delta\theta$ (filled symbols) and out-of-plane $\Delta\varphi$ (empty symbols) dynamics in NiFe averaged over the whole layer thickness, during one period of oscillation. The $\pi/2$ phase difference in the sinusoidal fittings accounts for the right-hand precession. Scale bars 100 nm. In (**a**) and (**c**) the dynamics is enhanced by a factor of 5 for better visualization.

CoFeB. In the case of the out-of-plane components, the localization at the interface can be ascribed to the dynamic dipolar field tendency to close within the Ru spacer rather than outside of the sample. Noteworthy, this is a different mechanism with respect to the surface localization of Damon-Eschbach waves, which is sizeable only when the spin-wave wavelength is comparable with the film thickness, and would cause both the in-plane and out-of-plane dynamics to be maximized in the middle of the structure[36], in contrast with our observations. For comparison, the simulated amplitude profiles of spin waves in a compensated SAF are shown in Supplementary Fig. 6 and as expected do not show any strong depth-dependence of the in-plane amplitude profile.

These observations suggest that through materials engineering it is possible to generate complex three-dimensional features in the spin-wave modes which, as discussed in the following, have major effects on their propagation and interaction. Towards applications, controlling the vertical localization of spin waves is vital to the realization of three-dimensional magnonic networks for the parallel propagation of multiple spin-wave modes. In these systems, as spin-wave propagate and spatially superimpose through the thickness, they are expected to generate depth-dependent interference patterns, which would be key to spin-wave based processing. However, to our knowledge, three-dimensional interference has not been observed in spin waves so far,

and the possible mechanisms of its generation have not been investigated.

To study the three-dimensional structure of spin-wave interference, we focus on a region (see Fig. 1f) where two spin-wave wavefronts, propagating at an angle of ~50° with respect to each other, spatially superimpose and interfere as shown in the top view of Fig. 4a. Examples of time-resolved STXM highlighting the interference region are shown in Supplementary Fig. 7.

By reconstructing a time-resolved snapshot of the in-plane dynamics in the spin-wave interference region, we observe the emergence of intrinsically three-dimensional features within the volume, which are remarkably different in the two layers. In particular, the interference pattern (Fig. 4c) in NiFe displays steep vertical features, identified by the $\Delta\theta = 0$ surface where the in-plane dynamics vanishes, whereas in CoFeB we identify a complex configuration, characterized by a saddle-shaped surface of ~$150 \times 150$ nm$^2$ located in the center of the CoFeB layer. Interestingly, we find that the three-dimensional saddle structure corresponds to a region where both the in-plane dynamics $\Delta\varphi$ and the out-of-plane dynamics $\Delta\varphi$ (Supplementary Figs. 8–11) vanish, giving rise to a destructive interference pattern localized within the volume of the system. By extracting horizontal sections at different depths (Fig. 4d), we observe that the in-plane dynamics exhibit opposite sign above and below the saddle (see also

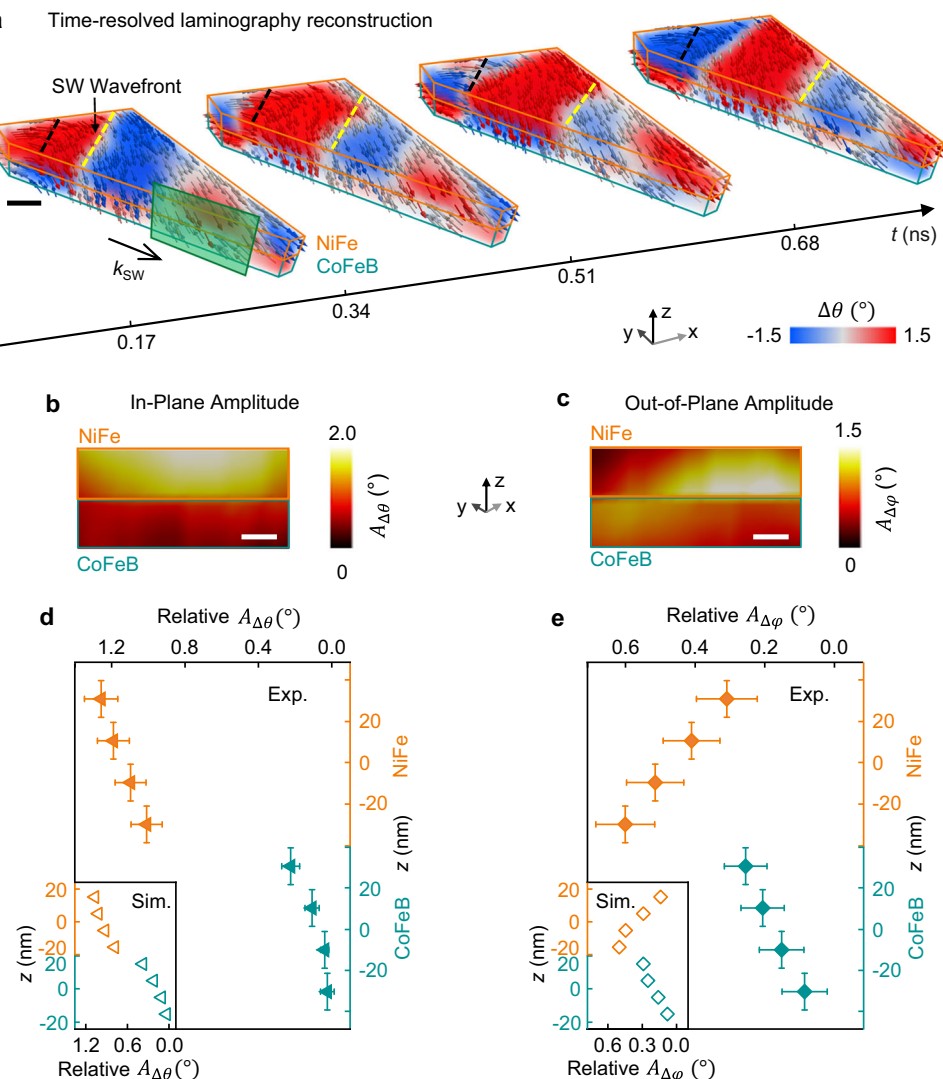

**Fig. 3 | Mapping the spin-wave amplitude and localization through the films thickness. a** The sequence of time-resolved reconstructions shows the propagation of planar spin-wave wavefronts (yellow line where $\Delta\theta = 0$) emitted by a domain wall (black line). Scale bar, 150 nm. **b**, **c** The amplitude of the in-plane ($A_{\Delta\theta}$, **b**) and out-of-plane ($A_{\Delta\varphi}$, **c**) dynamics are mapped through the thickness and along the propagation direction, in correspondence of the green cross section of panel (**a**). Both display complex, non-uniform profiles through the thickness. Scale bar, 50 nm. **d**, **e** By plotting the relative variation with respect to the bottom surface of the in-plane ($A_{\Delta\theta}$, **d**) and out-of-plane amplitude ($A_{\Delta\varphi}$, **e**), a remarkably different depth-dependence is observed: the in-plane amplitude is maximum at the top NiFe surface and decreases monotonically towards the bottom, while the out-of-plane dynamics is maximum in the middle of the structure and decreases at the top and bottom surfaces. Each amplitude value was calculated by averaging over a 480 nm x 120 nm horizontal section oriented along the spin-wave propagation direction. Examples of time-traces and the corresponding sinusoidal fittings used for extracting the amplitude values are shown in Supplementary Fig. 5. The corresponding micromagnetic simulations are shown in the insets.

Supplementary Movie 9). Noteworthy, this feature results from the superposition of two spin-wave wavefronts in antiphase, consistent with destructive interference, and is not observed in the case of free-space propagation, where the dynamics has always the same sign throughout the thickness.

To gain further insight on the dynamics within the interference, we performed micromagnetic simulations, shown in Fig. 4e, f (see also Supplementary Fig. 10-11 and Movie 10), and developed a general model for calculating and visualizing in three dimensions the interference pattern generated by two planar sinusoidal waves with $z$-dependent amplitudes, which nicely reproduce the three-dimensional pattern we observe experimentally. We find that the saddle-shaped $\Delta\theta = 0$ surface in CoFeB is a propagating dynamic structure, located in correspondence of a "buried" destructive interference region, where

the spin-wave amplitude is zero (more details in Supplementary Fig. 12 and related discussion). This originates from the interference of spin waves with different depth-dependent dynamics, giving rise to a distribution of minima and maxima within the volume, and to three-dimensional features within the propagating wavefronts, such as our saddle-shape structure.

Remarkably, the emergence of such three-dimensional interference patterns ultimately originates from the different magnetic moments of the two layers of the SAF, as shown in details also in Fig. 3 and related discussions, and therefore can be controlled by proper tuning of the structure. These observations demonstrate the three-dimensional nature of spin-wave interference and indicate a route to generate and manipulate three-dimensional interference patterns. Given their fundamental origin, such features are expected to be

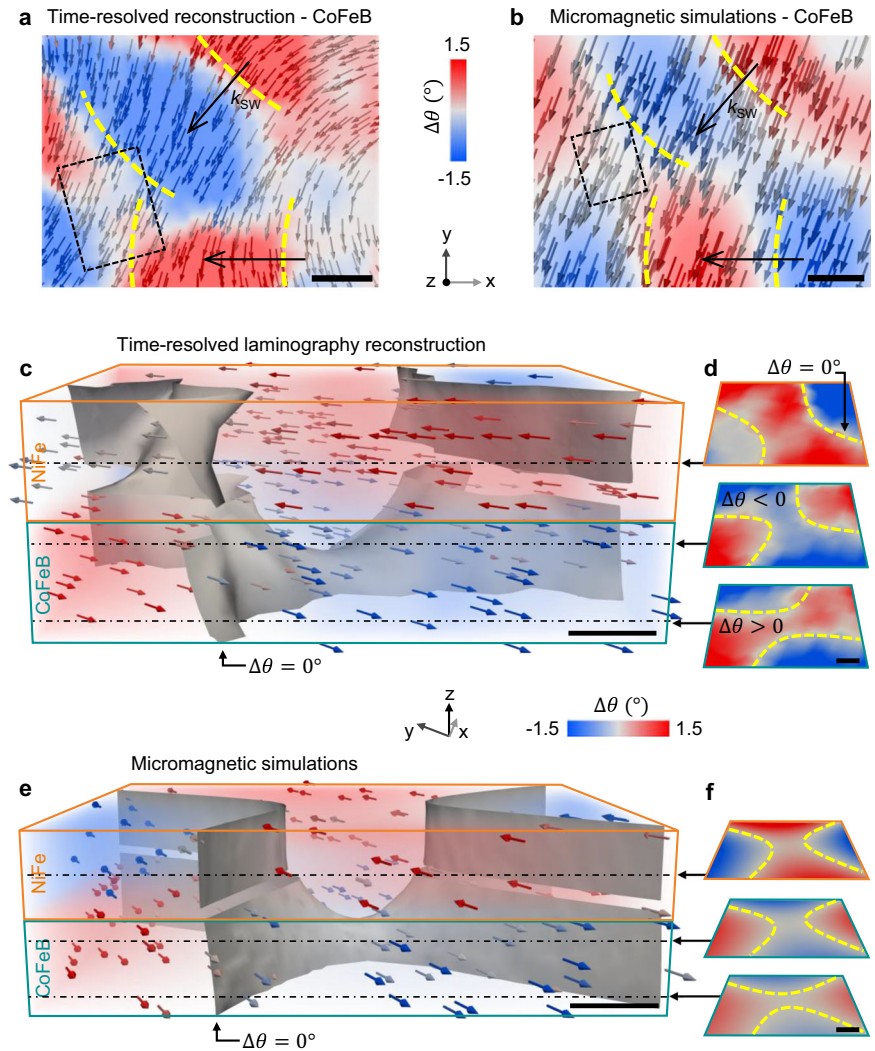

**Fig. 4 | Observation of three-dimensional spin-wave interference.**
**a**, **b** Experimental reconstruction (**a**) and corresponding micromagnetic simulations (**b**) showing two spin-wave wavefronts (yellow lines) propagating at an angle of 50° with respect to each other, and interfering destructively in correspondence of the rectangular region. Scale bar, 200 nm. **c** Experimental reconstruction of the interference in the rectangular region in (**a**). The gray surface, where the in-plane dynamic angle $\Delta\theta$ vanishes, reveals a complex three-dimensional saddle-shaped structure, corresponding to interference minima, located within the volume of the system. Scale bar, 50 nm. **d** Horizontal sections extracted from (**c**) at different heights reveals the sign change of the in-plane dynamics through the thickness, associated to the destructive interference of two waves in antiphase. Scale bar, 50 nm. **e**, **f** Micromagnetic simulations of the spin-wave interference region (**e**) and corresponding horizontal cross sections (**f**) to compare with (**c**) and (**d**) respectively. Scale bar, 50 nm.

hosted in a variety of magnetic systems, from single layers with surface-localized spin waves, to magnetic multilayers, to three-dimensional nanostructures, and represent an additional degree of freedom for designing novel functions in vertically integrated magnonic platforms.

In this work, we reconstructed the full three-dimensional landscape of coherent propagating spin waves. We showed experimentally that three-dimensionality governs some of the most fundamental aspects of their phenomenology, from their precessional dynamics to the localization and interference. This opens the way to investigate complex modes in thin films and heterostructures, and to study the interaction of spin waves with complex spin textures, buried defects, and non-uniform materials composition. Towards applications, our results open the path to characterize and control spin waves in three-dimensional nanostructures and magnonic networks, enabling the design of novel functions in next generation spin computing devices.

## Methods

### Microstructures and stripline fabrication

The synthetic antiferromagnet (SAF) microstructures described in this manuscript were fabricated by electron beam lithography followed by lift-off using a Vistec EBPG5000 100 kV electron beam writer. A bilayer resist composed of a layer of methyl methacrylate (6% dilution in ethyllactate, spincoated at 3000 rpm for 1 min) followed by a layer of poly(methyl metachrylate) (4% dilution in Anisole, spincoated at 3000 rpm for 1 min) was spincoated on top of the $Si_3N_4$ membrane. For both layers, a soft bake at 175 °C for 1 min was performed following the spincoating. The samples were exposed with a dose of 1500 $\mu C/cm^2$ and developed for 45 s by immersion in a 1:3 volume solution of methyl-isobutyl-ketone and isopropyl alcohol, followed by a 90 s immersion in pure isopropyl alcohol. A CoFeB (50) / Ru (0.5) / NiFe (40) / Ru (4) (thickness in nm) multilayer structure was grown via DC (Ru, NiFe) and RF (CoFeB) magnetron sputtering in an AJA Orion 8 System, with a base pressure below $1 \times 10^{-8}$ Torr[35,41]. The magnetic

characterization of the stacks was performed in a Microsense vibrating sample magnetometer.

Following the deposition of the SAF layers, the unexposed resist (and the magnetic film on top of it) was lifted off by immersion of the samples in pure acetone.

On top of the SAF microstructures, a 5 μm wide, 300 nm thick, Cu stripline was patterned by electron beam lithography followed by liftoff using the same recipe as for the SAF microstructures. The Cu film was deposited by thermal evaporation using a Balzers BAE250 evaporator.

## Micromagnetic simulations

Micromagnetic simulations were carried out by solving the Landau–Lifshitz–Gilbert equation of motion, using the open-source GPU-accelerated software MuMax3[42]. The material parameters used in the simulation were the following: saturation magnetization $M_s$ NiFe = 800 kA m$^{-1}$ and $M_s$ CoFeB = 1250 kA m$^{-1}$, exchange constant $A_{ex}$ NiFe = $0.75 \times 10^{-11}$ J m$^{-1}$ and $A_{ex}$ CoFeB = $1.2 \times 10^{-11}$ J m$^{-1}$, and interlayer exchange coupling constant J = $-1.2$ mJ m$^{-2}$. The Gilbert damping was set to $\alpha$ NiFe = 0.01 and $\alpha$ CoFeB = 0.008. Spin waves were excited by using narrow line excitations, where a sinusoidal out-of-plane magnetic field oscillating at a frequency of 0.86 GHz was applied. The magnitude of the excitation field was in the 5 mT – 7 mT range for all the simulations. In order to simulate the z-dependence of the SW amplitudes (Fig. 3d, e), a rectangular stripe geometry of 20.48 μm × 20 nm × 90 nm was employed. The total simulated volume was discretized into cells having dimensions of $5 \times 5 \times 5$ nm$^3$. In this case, spin waves were excited by using a single narrow line excitation in the middle of the rectangular stripe. For simulating the 3D interference (Fig. 4b, e, f), a 2.5 μm × 2.5 μm × 90 nm volume with cell dimensions of $5 \times 5 \times 5$ nm$^3$ was employed. Spin waves were excited by using two single narrow line excitations positioned at a 50° angle between each other. Periodic boundary conditions within the plane were used in both cases.

## Time-Resolved Soft X-ray Laminography

The three-dimensional time-resolved images were acquired according to the laminographic imaging protocol[32,33,43] where two-dimensional time-resolved projections at different rotation angles of the sample with respect to the X-ray beam were acquired. Each projection was acquired by time-resolved scanning transmission X-ray microscopy (STXM) imaging, where the two-dimensional images are acquired by focusing a monochromatic X-ray beam with a diffractive optical element (Fresnel zone plate) on the surface of the sample, and recording the transmitted intensity with an avalanche photodiode detector. The combination of the outermost zone width of the Fresnel zone plate of 30 nm, and of the size of the secondary source selected for the experiments allows for the focusing of the X-ray beam to a spot size of less than the 40 nm step size used in the acquisition of the STXM images used for the laminographic reconstruction. To obtain the image, the sample is scanned with a piezoelectric scanner, with its position controlled by means of an interferometric sample positioning system[44]. To independently probe the two layers of the SAF investigated in this work, the X-ray energy was tuned to the L$_2$ absorption edges of Co (781 eV) and Ni (856 eV).

Due to mechanical constraints, a laminography angle of 45° was selected. This configuration allowed us to be sensitive both to the in-plane and out-of-plane components of the magnetization, even if in our SAF system the latter is much smaller, resulting in a lower signal-to-noise-ratio.

A total of 37 projections were acquired at both the Co and Ni L$_2$ edges, with the angular sampling corresponding to a maximum z-resolution of 10 nm[32,33,43]. In general, for the laminography measurements, $N_P$ projections measured over 360° are required to obtain a spatial resolution $\Delta r$, defined as $N_P = (\pi * t / \triangle r) * \tan \theta_L$, where $t$ is the thickness of the sample and $\theta_L$ is the laminography angle. Following this equation, for this reconstruction the angular sampling corresponds to a nominal spatial resolution $\triangle r$ of ~ 10 nm. It is worth noting that, in reality, the experimental spatial resolution is poorer than the nominal one, due to the optics constraints and signal to noise ratio of the XMCD projections. To align the projections used for the reconstruction of the two layers a sub-pixel registration procedure is performed before the reconstruction, based on the topographical features of the SAF microstructure[45]. For each of the 37 projections, two time-resolved STXM images were acquired using circularly polarized X-rays of opposite helicities. This is necessary to allow for the full reconstruction of the three-dimensional orientation of the magnetization vectors[32,33]. The time-resolved images consist of a set of 7 frames for each projection, where the detection is performed by a combination of a fast avalanche photodiode and a dedicated field-programmable gate array that handles the temporal sorting of the recorded photon counts, as described in detail in ref. 44 The frequency of the RF signal used to excite the spin waves was selected according to the relation $f = 500 M / N$ MHz, where $N = 7$ is equal to the number of frames in the image, and $M$ is an integer not multiple of $N$[44]. For the work described in this manuscript, $M$ was selected to be equal to 12, yielding an excitation frequency of 0.86 GHz. The time step of the time-resolved images is equal to $2/M$ ns, i.e. about 160 ps, and the temporal resolution of the scans is given by the width of the X-ray pulses generated by the synchrotron light source, which is 70 ps FWHM[46].

The RF excitation was generated by a Keysight M8195A arbitrary waveform generator with a 64 GSa/s bandwidth frequency locked to the 500 MHz master clock of the synchrotron light source. To obtain the 2 V amplitude (monitored with a 50 Ω terminated oscilloscope at the injection point) used to drive the spin wave generation, we employed a mini-circuits ZHL-4240W amplifier.

To contact the stripline used to generate the oscillating magnetic field that drives the spin-wave excitation, a custom-designed printed circuit board (PCB), designed to be compatible with the constraints of the laminography stage while still guaranteeing good RF performances, was used[33].

## Laminography reconstruction and data analysis

The three-dimensional topographic and magnetic configuration reconstruction of the SAF microstructure was performed independently for each of the 7 frames of the time-resolved images acquired at the Co and Ni edges according to the algorithm described in refs. 32,47. Prior to the magnetic reconstruction, each projection was upsampled with a 2x factor using a bicubic interpolation. With this, a 3D magnetic reconstruction with a $20 \times 20$ x 20 nm$^3$ voxel size was obtained, analogous to the procedure demonstrated in refs. 48–50. The output of the reconstruction algorithm was the three-dimensional magnetization vector field, $M_x(t)$, $M_y(t)$, $M_z(t)$, for the CoFeB and NiFe layer, separately, for each of the 7 frames. The static magnetization vector was obtained by averaging each component of the magnetization over the 7 frames ($<M_x>$, $<M_y>$, $<M_z>$). The magnetization dynamics was obtained for each component $i$ as $\Delta M_i(t) = M_i(t) - <M_i>$, from which the in-plane dynamic angle $\Delta \varphi(t)$ and the out-of-plane dynamic angle $\Delta \theta(t)$ were calculated trigonometrically. The in-plane and out-of-plane spin-wave amplitudes $A\Delta\theta$ and $A\Delta\varphi$ are defined as the amplitudes of the sinusoidal fit of the time-traces of $\Delta\theta$ and $\Delta\varphi$.

For extracting the SWs spatial profiles and time-traces shown in the graphs of Fig. 2 and Fig. 3 the following procedures were employed:

Figure 2d: The experimental SW spatial profiles of the in-plane dynamic angle $\Delta\theta$ propagating towards -x for both NiFe and CoFeB were extracted from the yellow plane (for the whole thickness of each

layer). The obtained data have then been averaged, giving the reported results. The error bars indicate the standard deviation of the values.

Figure 2e: The time-traces of the in-plane and out-of-plane dynamic angles $\Delta\theta$ and $\Delta\varphi$ for the NiFe layer were extracted from the yellow plane shown in Fig. 2b at a distance of ~ 200 nm from the right border. The obtained data have then been averaged, giving the reported results. The error bars indicate the standard deviation of the values.

Figure 3d, e: The variations of the in-plane SW amplitude $A_{\Delta\theta}$ and out-of-plane SW amplitude $A\Delta_\varphi$ for the NiFe and CoFeB layers were calculated from the reconstructed region of Fig. 3a. The amplitudes were calculated by fitting the $\Delta\theta$ $(t)$ and $\Delta\varphi$ $(t)$ values with a sine function. The $z$-dependence of the amplitude was obtained by averaging over a 480 nm x 120 nm rectangular horizontal section oriented along the spin-wave propagation direction for each $z$-value (See Supplementary Fig. 5). Then, the variation was calculated as the difference between the amplitudes and their value at the bottom CoFeB surface. The corresponding error bars indicate the averaged standard deviation of each wavefront.

## Data availability
The raw synchrotron data, reconstruction data and simulations data generated in this study have been deposited in the Zenodo database under accession code 10.5281/zenodo.10814009.

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

## Acknowledgements

This work was partially performed at PoliFAB, the microtechnology and nanotechnology centre of the Politecnico di Milano. The research leading to these results has received funding from the European Union's Horizon 2020 research and innovation program under grant agreement number 948225 (project B3YOND). All synchrotron data was measured at the PolLux endstation of the Swiss Light Source, Paul Scherrer Institut, Villigen PSI, Switzerland. The PolLux endstation was financed by the German Bundesministerium für Bildung und Forschung (BMBF) through contracts 05K16WED and 05K19WE2. E.A. acknowledges funding from the European Union – Next Generation EU – "PNRR – M4C2, investimento 1.1 – "Fondo PRIN 2022" – TEEPHANY–ThreEE-dimensional Processing tecHnique of mAgNetic crYstals for magnonics and nanomagnetism ID 2022P4485M CUP D53D23001400001", and from the FARE programme of the Italian Ministry for University and Research (MUR) under grant agreement R20FC3PX8R (project NAMASTE). D.P. acknowledges funding from the European Union – Next Generation EU – "PNRR – M4C2, investimento 1.1 – "Fondo PRIN 2022" – PATH – Patterning of Antiferromagnets for THz operation id 2022ZRLA8F – CUP D53D23002490006" and from Fondazione Cariplo and Fondazione CDP, grant n° 2022-1882. S.M. acknowledges funding from the Swiss National Science Foundation (Grant Agreement 172517). C.D. acknowledges funding from the Max Planck Society Lise Meitner Excellence Program and funding from the European Research Council (ERC) under the ERC Starting Grant 3DNANOQUANT 101116043.

## Author contributions

E.A., S.F. and D.P. conceived the study and designed the experiments. D.G., S.F., S.C., F.M., D.P. and E.A. fabricated the samples. D.G., S.F., G.R. and S.M. performed the synchrotron experiments with the support of C.D., D.P. and E.A. D.G. and S.F. performed the laminography reconstruction with the support of C.D., D.P. and E.A., from a code provided by S.F. and C.D. D.G., D.P. and E.A. performed the data analysis with the support of S.F. and C.D. D.G. and E.A. performed the micromagnetic simulations. D.G., D.P. and E.A. wrote the manuscript with contributions from all authors. All authors contributed to the discussion and interpretation of the results.

## Competing interests

The authors declare no competing interests.
