## [Peer Review File · Nature Communications]

Reviewers' Comments:

Reviewer #1:

Remarks to the Author:

Girardi et al report on 3D spin wave profiles in a magnetic NiFe/Ru/CoFeB trilayer with antiferromagnetic coupling between the layers. Magnetic laminography is used to obtain 3D vector magnetization profiles with high spatial and temporal resolution. This allows the authors to probe SW dynamics in the different layers and to observe non uniform SW profiles across the thickness due to the competition between AF exchange and dipolar coupling.

This work is part of a research line by this group on the development of advanced vector imaging techniques for magnetization dynamics, which are relevant in the field of magnonics. In previous works, they have already studied spin waves in single magnetic films and the motion of a magnetic vortex core in a microdisk. The results presented here are a qualitative step forward since they observe for the first time the 3D spin precession in spin waves and offer novel experimental insight into the depth dependence of spin wave dynamics in multilayers. Thus, I consider that this work can be very relevant in the field of magnonics as it can provide an accurate and reliable experimental technique for the characterization of 3D SW profiles.

For these reasons, I consider that it is very important that the authors take an additional effort to assess the accuracy of the reported 3D SW profiles and the actual spatial and temporal resolution in each of the experimental cases reported in the manuscript.

Briefly, the characterization of spin waves propagating in a magnetic microstructure requires the determination of the angular deviations of the magnetic moment from its average position as a function of time and position. Here, the authors present their results in a sequential manner: starting with time and spatial dependence of the deviation angles ($\Delta\theta$ and $\Delta\phi$ in figure 2), following with the spatial dependence of oscillation amplitudes in figure 3 and, finally, presenting 3D interference patterns based on the analysis of $\Delta\theta$ in figure 4. My concern is that averaging procedures applied to reported data are different in each of these three cases so that the reported experimental accuracy in e. g. the data in Figure 2 might not be directly applicable to the data in Figure 4.

In particular, the following points need to be addressed in order to improve the clarity of the manuscript:

1) Figure 1 describes the sample and the experimental set-up, with a strip line antenna that creates a rf-field on the magnetic microstructure. SW propagation directions are sketched by black arrows in Fig. 1d. How are these k_{sw} determined? How are they related to the excitation field geometry or to the different spin textures in the microstructure? Which is the lateral resolution of STXM images and for the laminography reconstruction? What is the procedure (and accuracy) for the spatial alignment between the reconstructed magnetization data in the NiFe and CoFeB layers?

2) Figure 2 shows the time and spatial dependence of $\Delta\theta$ and $\Delta\phi$, however part of the figure caption

regarding the size of averaging regions in Figs 2d-e is placed in the Methods section. The reported $\Delta\theta(t,x)$ and $\Delta\phi(t)$ profiles correspond to an average over the full layer thickness so that this information should be included in the caption. Also, the yellow line in Fig. 2b that marks the studied region should be replaced by a rectangle, indicating that the information is coming from the full layer thickness.

3) Figure 3 shows the depth and lateral dependence of the SW oscillation amplitudes. Which is pixel size in Figs. 3b-c? What is the size of the averaging region used to obtain the amplitude values in each of these pixels? In fig. 3a, yellow lines indicate a propagating wavefront (is it defined as a line for zero $\Delta\theta$?). Which is the definition of the 15 wavefronts used to perform the average needed to obtain the data in Fig. 3 e-d, as described in the methods section? In general, the size of the averaging region used to obtain each data point should be clearly stated in the figure caption. Also, it would be advisable to include several of the $\Delta\theta(t)$ and $\Delta\phi(t)$ curves used to construct this figure in the supplementary information.

4) Figure 4 shows a snapshot of the time resolved reconstruction and the shape of the $\Delta\theta=0$ surface is analyzed in terms of the destructive interference between propagating SW wavefronts. It would be good if the authors could include a direct STXM image in which the changes in contrast related with the change in sign of $\Delta\theta$ could be directly appreciated (even if it is a 2D projection). Which is the voxel size used to determine $\Delta\theta$ values in this figure? What is the lateral and vertical resolution in the determination of the $\Delta\theta=0$ surface taking into account the noise level in $\Delta\theta$?

5) The nature of the "destructive interference pattern" between SWs should be explained more clearly. Usually, an interference pattern indicates the presence of specific spatial points in which a constant π phase difference results in zero oscillation amplitude (i.e. a static $\Delta\theta=0$ surface). Here, the spatial location of the $\Delta\theta=0$ surface moves in consecutive snapshots both in the experiment and the micromagnetic simulations, as corresponds to a moving wavefront rather than to an interference pattern. Also, they should explain more clearly the nature of "transient dynamics" in the context of a periodic excitation of constant amplitude. Their micromagnetic simulations only cover a fraction of the temporal period of the rf field. How is the evolution of the $\Delta\theta=0$ surface over the full excitation period?

In summary, I consider that this work reports on a very interesting advance of vector imaging of spin waves with good spatial and temporal resolution, revealing the complex 3D configuration of magnons in magnetic multilayers that had not been reported before. In my opinion, this work will deserve publication in Nature Communications, once that the points described above are properly clarified.

Reviewer #2:

Remarks to the Author:

Dear editor and authors, please find in the following my review of the current state of the manuscript, starting with a short description of its content and findings:

In the presented manuscript, the authors showcase the magnetization imaging technique 'Time-Resolved Soft X-ray Laminography' on spin-wave dynamics in microstructures of a synthetic antiferromagnet.

For adequate samples, mainly thin enough to be sufficiently x-ray transparent, a rotation about an oblique axis (here taken for 37 angles at 45 degree tilt) allows capturing the magnetization with claimed nanoscale spatial resolution and sub-ns (70ps,160ps) temporal resolution. Furthermore, selecting photon-energies matching a materials L2 edge allows element specific signal collection.

The technique has been demonstrated previously in [1], [2] and utilized on magnetization dynamics of a domain wall and vortex gyration [3]. It bears great similarities to soft x-ray tomography techniques able to resolve the full magnetization vector field.

Despite its previous principled demonstration on magnetization dynamics it is in my opinion an indeed interesting and important progression to apply this imaging technique to other related magnetization dynamics, such as that for spin waves, in this case on synthetic antiferromagnets.

In particular as the spin-wave dynamics are substantially modified by coupling between layers and their local amplitude distribution should give insights into local interfacial torques , inhomogeneities, defects etc.

The work addresses these dynamics in rectangular microstructures of the synthetic antiferromagnet CoFeB(50nm)/Ru(0.5nm)/Ni(40nm). Large changes in the ellipticity of precession across the film thickness, and of the precession amplitude in particular near the common spacer-interface are recorded.

This is also expected based on the current understanding and micromagnetic simulations. A good agreement with these micromagnetic simulation and experimental results is observed throughout the addressed regions of the sample. The data is clearly presented and appears of high quality.

In my opinion the manuscript would greatly benefit from addressing the differences from the idealized micromagnetic simulation in particular on the -x,-y quadrant, where the expected Landau pattern is heavily distorted. The authors suggest variations of the interlayer exchange coupling to be the reason. Such differences should also change the spin-wave mode profile considerably, allowing to characterize these differences further based on the experimental data in tandem with micromagnetic simulations of different coupling constants. However, I do not view inclusion of this as a necessity for publication.

In the current state, the manuscript successfully demonstrates a high degree of agreement of the recorded magnetization dynamics with theoretical expectations and micromagnetic simulation such as general propagation and complex interference patterns.

This confirms the capability of the measurement technique to capture all dynamics accurately and the match with the simulation corroborates this as it is based on the previous demonstrations of the technique and other studies on spin waves in synthetic antiferromagnets.

In this respect, I recommend the work for publication with a few minor points to address:

Description of the micromagnetic simulations:

I would recommend to clarify the micromagnetic simulations for the different regions in particular region 4. The type of excitation and free propagation path prior to the observed spin-wave interference would be illuminating to the readers.

SI Figure 8: the color bar appears saturated making it impossible to read out the profile in the saturated regions from the given images. As the saturated regions take up more than 90% of the image, it would be certainly beneficial to find a solution. Either via a second separate color bar or logarithmic plotting etc.

SI Figure 6: The caption should be rephrased in my opinion as it currently reads a bit confusingly, but is technically fine. In b) the title should read $A(\Delta\phi)$ if I understand the caption correctly.

SI Figure 5: In my opinion this figure nicely clarifies, the extracted line profiles and could be moved up as additional information added to figure 1. Alternatively, the extraction region e.g. of figure 2 could/should be visualized there for clarification in my opinion.

As mentioned above, I believe the manuscript would greatly benefit from studying the distorted regions of the structure in regard of the information that can be gained by analyzing the local spin-wave precessions to infer on the type of defect/coupling change/ etc in those regions. However, as mentioned I do not view this as a necessity, rather a suggestion and believe that such imaging studies are of high value to the field of magnonics and its development into 3D structures.

[1] C. Donnelly et al., "Time-resolved imaging of three-dimensional nanoscale magnetization dynamics," *Nat. Nanotechnol.*, vol. 15, no. 5, pp. 356–360, May 2020, doi: 10.1038/s41565-020-0649-x.

[2] K. Witte et al., "From 2D STXM to 3D Imaging: Soft X-ray Laminography of Thin Specimens," *Nano Lett.*, vol. 20, no. 2, pp. 1305–1314, Feb. 2020, doi: 10.1021/acs.nanolett.9b04782.

[3] S. Finizio, C. Donnelly, S. Mayr, A. Hrabec, and J. Raabe, "Three-Dimensional Vortex Gyration Dynamics Unraveled by Time-Resolved Soft X-ray Laminography with Freely Selectable Excitation Frequencies," *Nano Lett.*, vol. 22, no. 5, pp. 1971–1977, Mar. 2022, doi: 10.1021/acs.nanolett.1c04662.

REPLY TO REVIEWERS

Reviewer #1:

Girardi et al report on 3D spin wave profiles in a magnetic NiFe/Ru/CoFeB trilayer with antiferromagnetic coupling between the layers. Magnetic laminography is used to obtain 3D vector magnetization profiles with high spatial and temporal resolution. This allows the authors to probe SW dynamics in the different layers and to observe non uniform SW profiles across the thickness due to the competition between AF exchange and dipolar coupling.

This work is part of a research line by this group on the development of advanced vector imaging techniques for magnetization dynamics, which are relevant in the field of magnonics. In previous works, they have already studied spin waves in single magnetic films and the motion of a magnetic vortex core in a microdisk. The results presented here are a qualitative step forward since they observe for the first time the 3D spin precession in spin waves and offer novel experimental insight into the depth dependence of spin wave dynamics in multilayers. Thus, I consider that this work can be very relevant in the field of magnonics as it can provide an accurate and reliable experimental technique for the characterization of 3D SW profiles.

For these reasons, I consider that it is very important that the authors take an additional effort to assess the accuracy of the reported 3D SW profiles and the actual spatial and temporal resolution in each of the experimental cases reported in the manuscript.

Briefly, the characterization of spin waves propagating in a magnetic microstructure requires the determination of the angular deviations of the magnetic moment from its average position as a function of time and position. Here, the authors present their results in a sequential manner: starting with time and spatial dependence of the deviation angles ($\Delta\theta$ and $\Delta\phi$ in figure 2), following with the spatial dependence of oscillation amplitudes in figure 3 and, finally, presenting 3D interference patterns based on the analysis of $\Delta\theta$ in figure 4. My concern is that averaging procedures applied to reported data are different in each of these three cases so that the reported experimental accuracy in e. g. the data in Figure 2 might not be directly applicable to the data in Figure 4.

We thank the reviewer for their appreciation of our work and for their review, which gave us the opportunity to clarify some technical aspects of our work. In addition, especially to address the referee's question on the achievable spatial resolution of the technique, which is a multi-faceted topic, we have performed additional analysis of the acquired data. As detailed in our answer to comment 1.10, we performed an upsampling of the acquired experimental data to obtain a 3D reconstruction with a voxel size closer to the theoretical limit of the achievable spatial resolution. The results obtained via 3D imaging are confirmed by micromagnetic simulations. Below a point-by-point reply to the questions is reported:

In particular, the following points need to be addressed in order to improve the clarity of the manuscript:

Comment 1.1: *1) Figure 1 describes the sample and the experimental set-up, with a strip line antenna that creates a rf-field on the magnetic microstructure. SW propagation directions are sketched by black arrows in Fig. 1d. How are these k_{sw} determined? How are they related to the excitation field geometry or to the different spin textures in the microstructure?*

The sketch represented in Fig. 1d corresponds specifically to acoustic spin waves propagating in a synthetic antiferromagnetic trilayer. These modes, as first predicted by the pioneering theoretical study performed by P. Grünberg in 1981¹ and later on experimentally²⁻⁴, are characterized by a non-reciprocal propagation, as a consequence of the dipolar coupling between the two ferromagnetic layers. This strong non-reciprocity causes the two branches $\pm k$ in the spin-wave dispersion relation reported in Extended Data Figure 3 to be significantly different. At the frequency characteristic of our experiment, only the $k > 0$ branch is excited, whose direction of propagation can be inferred from the right hand rule for $(\mathbf{M}_{top}, \mathbf{M}_{bot}, \mathbf{k}_{sw})^2$ and as reported below for our sketch:

In particular, the propagation direction k_{sw} can be estimated point-by-point in Fig. 1f and the corresponding Movie S5 and Movie S6, and is perpendicular to the SW wavefronts.

Similarly to⁴, the spin-wave excitation mechanism employed in this work is based on the coupling between the RF Oersted magnetic field generated by the stripline with the nanoscale spin textures (i.e., domain walls and vortex). The RF magnetic field generated by the stripline is characterized both by in-plane and out-of-plane components. The effect of such field is to set into oscillation vortices and domain walls, or more in general, non-uniform spin textures. For doing so the out-of-plane field couples with the out-of-plane component of the magnetization resulting in non-uniform textures, while the in-plane field acts on the “net” in-plane component of \mathbf{M} , resulting in non-compensated SAF. Detailed discussion is reported in^{2,4}. The symmetry and shape of the spin-wave wavefronts is closely related to the type, symmetry and shape of the oscillating textures: e.g. radial wavefronts are emitted from vortices and planar wavefronts are emitted from straight domain walls.

To further clarify these matters, we have included the following updates:

Manuscript main text - new updated version: To excite spin waves, a Cu radiofrequency antenna was fabricated on top of the structure (Fig. 1a). **The coupling between the RF Oersted magnetic field generated by the stripline and the spin textures in the microstructure allows for the efficient generation of spin waves. Additionally, in SAFs, the interlayer exchange and dipolar interaction gives rise to coupled nonreciprocal spin-wave modes in the two layers. In particular, at our experimental excitation frequency of 0.86 GHz, chosen to maximize the spin-wave excitation efficiency, we expect to have only acoustic spin waves, characterized by antiparallel in-plane / parallel out-of-plane magnetization dynamics^{32,36}. Such nonreciprocal acoustic modes are sketched in Figure 1d with k_{sw} indicating the propagation direction. The strong nonreciprocity of the investigated SAF is further analyzed by the simulated dispersion relation shown in Extended Data Figure 3, and confirms the unidirectionality of the spin-wave propagation at our experimental excitation frequency.**

A snapshot of the three-dimensional reconstruction of the magnetization dynamics in the whole sample is shown in Fig. 1f, where the arrows indicate the magnetization direction across the volume, and the color code shows the in-plane dynamic angle $\Delta\theta$, **associated to the spin waves propagating along k_{sw} .**

Manuscript Figure 1 caption - new updated version: [...] **d**, Schematic of **nonreciprocal** acoustic spin waves propagating in a synthetic antiferromagnet with $k>0$. [...]

Extended Data Fig. 3 and caption - new updated version:

Extended Data Figure 3 | Micromagnetic simulation of the acoustic SW mode dispersion in the SAF. **a**, The magnetization orientation in the top NiFe layer and the propagation direction in both the positive (+k, red) and negative (-k, blue) branches of the dispersion is highlighted. The green point represents the experimental data point. **b**, Right-hand rule for (M_{top} , M_{bot} , k_{sw}) showing the direction of propagation for the $k > 0$ branch of the acoustic modes.

Supplementary Information Section 3 text - new version: The strong non-reciprocity of the SAF, is confirmed by observing the +k (red) and -k (blue) branches^{1,2}. The green point represents our experimental data point ($f = 0.86$ GHz, $k = 0.97 \pm 0.07 \cdot 10^7$ rad/m); it is worth to notice that at the excitation frequency of our experiment, the propagation of the mode is unidirectional (see Extended Data Figure 3b), resulting in the absence of back-reflection from the boundaries or defects.

C1.2: Which is the lateral resolution of STXM images and for the laminography reconstruction?

The spatial resolution of the STXM technique depends on the combination of the X-ray beam spot, but also on the step size of the piezoelectric scanner utilized during the acquisition of the image. In the case presented in this manuscript, the combination of the size of the secondary source and of the 30 nm Fresnel zone plate used to focus the X-ray beam, leads to an X-ray spot size smaller than the pixel size used in the images (spot size ca. 30-35 nm, with a pixel size of 40 nm), indicating that the images are primarily limited by the step size utilized for the scans. The reason for this choice was to allow for the acquisition of the 74 sets of time-resolved images (37 projections with 2 helicities per projection) within the allocated beamtime. Regarding the resolution of the laminography reconstruction, please refer to comment C1.10 down below.

This information has been explicitly added to the methods section:

Each projection was acquired by time-resolved scanning transmission X-ray microscopy (STXM) imaging, where the two-dimensional images are acquired by focusing a monochromatic X-ray beam with a diffractive optical element (Fresnel zone plate) on the surface of the sample, and recording the transmitted intensity with an avalanche photodiode detector. The combination of the outermost zone width of the Fresnel zone plate of 30 nm, and of the size of the secondary source selected for the experiments allows for the focusing of the X-ray beam to a spot size of less than the 40 nm step size used in the acquisition of the STXM images used for the laminographic reconstruction.

C1.3: *What is the procedure (and accuracy) for the spatial alignment between the reconstructed magnetization data in the NiFe and CoFeB layers?*

We thank the reviewer for the insightful question and for the opportunity to further clarify this matter. In our case, it is important to emphasize that thanks to the elemental sensitivity of STXM, the two layers of the SAF (NiFe and CoFeB) were probed and reconstructed independently, by tuning the X-ray energy at the L2 absorption edges of Co (781 eV) and Ni (856 eV), and acquiring multiple projections at different rotation angles. At each of these rotation angles, first the positively circularly polarized (C+) and negatively circularly polarized (C-) measurements were taken at the Ni-edge, and immediately after at the Co-edge. For this reason, only a small drift between the XMCD images of the two layers could occur, comparable with the step size of 40 nm employed in the measurements. To correct for such drift, during the first step of the reconstruction algorithm, an alignment procedure which exploits the topographical feature of the SAF microstructure is performed.

Such alignment procedure and its accuracy is described in detail in [Odstrčil, M. et al. Alignment methods for nanotomography with deep subpixel accuracy. *Opt. Express* 27, 36637–36652 (2019).], and has been used in previous Laminography experiments, such as in [C. Donnelly et al., “Time-resolved imaging of three-dimensional nanoscale magnetization dynamics” *Nat. Nanotechnol.*, vol. 15, no. 5, pp. 356–360, May 2020, doi: 10.1038/s41565-020-0649-x] and [K. Witte et al., “From 2D STXM to 3D Imaging: Soft X-ray Laminography of Thin Specimens” *Nano Lett.*, vol. 20, no. 2, pp. 1305–1314, Feb. 2020, doi: 10.1021/acs.nanolett.9b04782].

To clarify this point we added a sentence to the methods section:

To align the projections used for the reconstruction of the two layers a sub-pixel registration procedure is performed before the reconstruction, based on the topographical features of the SAF microstructure [Odstrčil, M. et al. Alignment methods for nanotomography with deep subpixel accuracy. *Opt. Express* 27, 36637–36652 (2019).].

C1.4: 2) *Figure 2 shows the time and spatial dependence of $\Delta\theta$ and $\Delta\phi$, however part of the figure caption regarding the size of averaging regions in Figs 2d-e is placed in the Methods section. The reported $\Delta\theta(t,x)$ and $\Delta\phi(t)$ profiles correspond to an average over the full layer thickness so that this information should be included in the caption. Also, the yellow line in Fig. 2b that marks the studied region should be replaced by a rectangle, indicating that the information is coming from the full layer thickness.*

As the reviewer suggested, we have updated Figure 2 accordingly.

Figure 2 | Three-dimensional dynamics of the acoustic spin-wave mode. a-c, In this region of the sample (b), coherent spin waves propagate freely along k_{SW} , akin to the schematic of Fig. 1d. By mapping the dynamics of all three components of the magnetization in NiFe (a) and CoFeB (c), the “right-hand” precessional motion (full arrows) around the static magnetization direction (black arrows), and the periodic spatial phase difference associated to coherent wave propagation, are directly observed. (See corresponding time-resolved Movies S2, S4 for the full dynamics). d, The antiparallel coupling of the in-plane dynamics $\Delta\theta$, signature of the acoustic modes, is highlighted by the **averaged** profiles extracted along the yellow plane in (b) **for the whole thickness of each layer**, from NiFe (orange) and CoFeB (blue), showing sinusoidal oscillations in antiphase, and a spin-wave wavelength of ~ 650 nm. e, Time-traces of the in-plane $\Delta\theta$ (filled symbols) and out-of-plane $\Delta\phi$ (empty symbols) dynamics in NiFe **averaged over the whole layer thickness**, during one period of oscillation. The $\pi/2$ phase difference in the sinusoidal fittings accounts for the right-hand precession. Scale bars, 100 nm (b), 50 nm (a, c). In (a) and (c), the dynamics is enhanced by a factor of 5 for better visualization.

C1.5: 3) Figure 3 shows the depth and lateral dependence of the SW oscillation amplitudes. Which is pixel size in Figs. 3b-c? What is the size of the averaging region used to obtain the amplitude values in each of these pixels?

The amplitude maps (calculated from the amplitudes of the sinusoidal fit of the time-traces of $\Delta\theta$ and $\Delta\phi$) of Fig. 3b,c do not show averaged values, but rather report the data point for the green slab of Fig. 3a, 1-pixel-thick in the plane of the sample of the first snapshot of Fig. 3a. A 20 nm pixel size was employed in all laminography images (see C1.10 for details) The slab extends through the full thickness of the sample. We modified the main text of the manuscript to better clarify this information:

To study the spin-wave properties through the thickness and along the propagation direction, we map the in-plane and out-of-plane spin-wave amplitudes $A_{\Delta\theta}$ and $A_{\Delta\phi}$, in correspondence of the

green vertical cross-section of the sample (see Methods). In particular, Fig. 3b,c report the data points corresponding to the green cross-section of Fig. 3a, visualized with a 20 nm x 20 nm pixel size in the horizontal and vertical direction.

C1.6: In fig. 3a, yellow lines indicate a propagating wavefront (is it defined as a line for zero $\Delta\theta$?).

Yes. We have added this information in the Figure caption (see comment C1.7).

C1.7: Which is the definition of the 15 wavefronts used to perform the average needed to obtain the data in Fig. 3 e-d, as described in the methods section? In general, the size of the averaging region used to obtain each data point should be clearly stated in the figure caption.

We thank the reviewer for this comment, as it is important to clarify the averaging procedure we used for the graphs in Fig. 3e,d.

First, the spin-wave amplitude for each voxel of the reconstructed microstructure was calculated by fitting the $\Delta\theta(t)$ and $\Delta\phi(t)$ with a sine function. Then, the amplitude values of each voxel in the rectangular horizontal 480 nm x 120 nm section located at different depths z (yellow rectangle below) were averaged. The horizontal sections were oriented perpendicular to the emitting domain wall, i.e., parallel to the k -vector of the emitted spin waves, so that the average was taken along the propagation path. For this reason, the term “wavefront” used in this context can generate some confusion and therefore we have removed the term and revised the Methods section.

To clarify these points the Methods section has been modified as follows:

Fig. 3d, e: The variations of the in-plane SW amplitude $A_{\Delta\theta}$ and out-of-plane SW amplitude $A_{\Delta\phi}$ for the NiFe and CoFeB layers were calculated from the reconstructed region of Fig. 3a. The amplitudes were calculated by fitting the $\Delta\theta(t)$ and $\Delta\phi(t)$ values with a sine function. The z -dependence of the amplitude was obtained by averaging over a 480 nm x 120 nm rectangular horizontal section oriented along the spin-wave propagation direction for each z -value (See Extended Data Fig. 5). Then, the variation was calculated as the difference between the amplitudes and their value at the bottom CoFeB surface. The corresponding error bars indicate the averaged standard deviation.

To underline the averaging procedure, we have also updated as suggested the figure caption as follows:

Figure 3 | Mapping the spin-wave amplitude and localization through the films thickness. a, The sequence of time-resolved reconstructions shows the propagation of planar spin-wave wavefronts (yellow line where $\Delta\theta = 0$) emitted by a domain wall (black line). Scale bar, 150 nm. **b, c,** The amplitude of the in-plane ($A_{\Delta\theta}$, **b**) and out-of-plane ($A_{\Delta\phi}$, **c**) dynamics are mapped through the thickness and along the propagation direction, in correspondence of the green cross section of panel a. Both display complex, non-uniform profiles through the thickness. Scale bar, 100 nm. **d, e,** By plotting the relative variation with respect to the bottom surface of the in-plane ($A_{\Delta\theta}$, **d**) and out-of-plane amplitude ($A_{\Delta\phi}$, **e**), a remarkably different depth-dependence is observed: the in-

plane amplitude is maximum at the top NiFe surface and decreases monotonically towards the bottom, while the out-of-plane dynamics is maximum in the middle of the structure and decreases at the top and bottom surfaces. Each amplitude value was calculated by averaging over a 480 nm x 120 nm horizontal section oriented along the spin-wave propagation direction. Examples of time-traces and the corresponding sinusoidal fittings used for extracting the amplitude values are shown in Extended Data Fig. 5. The corresponding micromagnetic simulations are shown in the insets.

C1.8: *Also, it would be advisable to include several of the $\Delta\theta(t)$ and $\Delta\phi(t)$ curves used to construct this figure in the supplementary information.*

We thank the reviewer for their suggestion. We have reported below as an example the time-trace and their sinusoidal fit of the in-plane and out-of-plane components of the two layers, extracted from 16 representative points of the averaging section (see answer above). The locations of the points is shown in panel b, below. It is possible to clearly recognize the two different trends of the amplitude profiles as a function of the z-value for the in-plane (black, on the left) and out-of-plane (red, on the right) components. Indeed, the in-plane $\Delta\theta(t)$ time-traces show a decreasing amplitude when going from the top NiFe surface towards the bottom CoFeB surface. On the other hand, the amplitude of the out-of-plane time-traces $\Delta\phi(t)$, present its maximum close to the interface between the two layers, in the middle of the structure.

We added Extended Data Fig. 5: “z-dependent spin-wave amplitude profiles”.

Extended Data Figure 5 | z-dependent spin-wave amplitude profiles. **a**, Snapshot of the time-resolved three-dimensional reconstruction of the region of the microstructure in which the z-dependent spin-wave amplitude profiles reported in Figure 3d,e of the main manuscript have been extracted. The amplitude values of each voxel in the yellow rectangular horizontal 480 nm x 120 nm section located at depth z were averaged. Scale bar, 150 nm. **b**, Same figure of panel a, highlighting the lines along which the time-traces of 8 points at different depth were analyzed. Scale bar, 150 nm. **c**, Time-traces of the in-plane $\Delta\theta$ (black symbols) and out-of-plane $\Delta\phi$ (red symbols) dynamics in NiFe and CoFeB, during one period of oscillation for the two different lines identified in (b).

In the main manuscript, in the caption of Fig. 3, we added: **Examples of time-traces and the corresponding sinusoidal fittings used for extracting the amplitude values are shown in Extended Data Fig. 5.**

C1.9: 4) *Figure 4 shows a snapshot of the time resolved reconstruction and the shape of the $\Delta\theta=0$ surface is analyzed in terms of the destructive interference between propagating SW wavefronts. It would be good if the authors could include a direct STXM image in which the changes in contrast related with the change in sign of $\Delta\theta$ could be directly appreciated (even if it is a 2D projection).*

We thank the reviewer for the suggestion. While it is not possible to observe directly from the STXM 2D projections the three-dimensional saddle-shape interference pattern, as it is buried inside the CoFeB layer, we have reported below an example of projection of the 7 frames taken over one period of excitation for the Ni and CoFeB layers, where the temporal evolution and the spin-wave propagation in correspondence of the interference region is visualized. As expected for a destructive interference region, a reduced contrast (grey color) is observed in all frames, both for CoFeB and NiFe. We added the following Extended Data Fig. 7 and caption to the Supplementary Information.

Extended Data Figure 7 | Examples of time-resolved laminography projections. Time-resolved STXM frames acquired for both NiFe and CoFeB at a projection angle of 316.8° and negative circular polarization, showing the interference region of Fig. 4 in the main text. In the

top (bottom) panel, the dynamics of the Ni (CoFeB) layer is visualized. As expected, the magnetic contrast associated to spin-wave propagation is reduced in the destructive interference region both in NiFe and CoFeB (indicated by the dashed box).

Main Manuscript: To study the three-dimensional structure of spin-wave interference, we focus on a region where two spin-wave wavefronts, propagating at an angle of $\sim 50^\circ$ with respect to each other, spatially superimpose and interfere as shown in the top view of Fig. 4a. **Examples of time-resolved STXM highlighting the interference region are shown in Extended Data Fig. 7.**

C1.10: *Which is the voxel size used to determine $\Delta\theta$ values in this figure? What is the lateral and vertical resolution in the determination of the $\Delta\theta=0$ surface taking into account the noise level in $\Delta\theta$?*

Regarding the spatial resolution, while it is challenging to provide an exact experimental value, it is possible to provide an estimate based on different aspects that affect that spatial resolution of the dataset. First, we can consider the angular resolution. For the three-dimensional imaging, the time-resolved Soft X-ray laminography set-up consisted of a rotation stage whose rotation axis is oriented at $\theta_L = 45^\circ$ with respect to the incident X-ray beam. A total of 37 complete projections were measured. In general, for the laminography measurements N projections measured over 360° are required to achieve a spatial resolution Δr defined as $N = (\pi * t / \Delta r) * \tan\theta_L$, where t is the thickness of the sample and θ_L is the laminography angle. Following this equation, for this reconstruction the angular sampling corresponds to a nominal spatial resolution Δr of ~ 10 nm. It is worth noting that in reality, the experimental spatial resolution is lower than the theoretical value given above, due to the geometric constraints of STXM imaging at soft X-ray energies combined with X-ray laminography [K. Witte *et al.*, Nano Letters 20, 1305 (2020)], which require to utilize a zoneplate with outermost zone width of 30 nm.

The single time-resolved STXM images obtained for each of the 37 angular projections were acquired with a pixel size of 40×40 nm² in xy. This relatively large pixel size is dictated by the stringent time limitations of synchrotron beamtimes, which is exacerbated by the fact that a TR-STXM image requires significantly higher statistics (i.e. integration time) compared to a static STXM image. However, given that both the beam spot size and the theoretical z resolution are better than 40 nm, we can obtain a 3D reconstruction with a smaller voxel size by upsampling the single TR-STXM images, prior to applying the reconstruction algorithm [Parker, J. A., Kenyon, R. V. & Troxel, D. E. Comparison of Interpolating Methods for Image Resampling. IEEE Transactions on Medical Imaging 2, 31–39 (1983); Mayerhoefer, M. E. et al. Effects of Magnetic Resonance Image Interpolation on the Results of Texture-Based Pattern Classification: A Phantom Study. Investigative Radiology 44, 405 (2009); Mahmoudzadeh, A. P. & Kashou, N. H. Evaluation of Interpolation Effects on Upsampling and Accuracy of Cost Functions-Based Optimized Automatic Image Registration. Int J Biomed Imaging 2013, 395915 (2013)]. In particular, we upsampled the XMCD projections to a pixel size approaching the angular resolution of the laminography scan, using an upsampling factor of 2x, which leads to a cubic voxel size of $20 \times 20 \times 20$ nm³. This is now presented in the revised version of the manuscript.

Importantly, for the dynamics that we see, such as the “saddle shape” $\Delta\theta = 0$ surface, and the smooth changes in the spin wave amplitudes through the thickness, the z dependence is relatively smooth and so a lower spatial resolution leads simply to a further smoothing of the spin-wave profiles, but does not prevent them from being identified. There remains some uncertainty as to the precise location of the measured position of the $\Delta\theta = 0$ surface. While it is extremely difficult to estimate quantitatively possible shifts in the measured location due to noise in the original STXM images, we performed extensive additional analysis, to investigate the origin of the saddle-shape structure. Following comment C1.11 by the reviewer, we developed from scratch a general model for calculating and visualizing the interference of sinusoidal propagating waves with depth dependent amplitudes. This allowed us to understand the full dynamics and origin of the saddle-shape structure (see C1.11 below), confirming that not only does it make sense that we observe the saddle point in our data, but that this is actually a feature one would expect to be present in all cases of interfering waves with three dimensional variations in amplitude.

In this framework, we made the following changes/additions to the main text, methods and supplementary:

Methods: A total of 37 projections were acquired at both the Co and Ni L_2 edges, with the angular sampling corresponding to a maximum z -resolution of 10 nm^{7,8,43}. In general, for the laminography measurements, N_p projections measured over 360° are required to obtain a spatial resolution Δr , defined as $N_p = (\pi * t / \Delta r) * \tan \theta_L$, where t is the thickness of the sample and θ_L is the laminography angle. Following this equation, for this reconstruction the angular sampling corresponds to a nominal spatial resolution Δr of ~10 nm. It is worth noting that, in reality, the experimental spatial resolution is poorer than the nominal one, due to the optics constraints and signal to noise ratio of the XMCD projections.

And:

The three-dimensional topographic and magnetic configuration reconstruction of the SAF microstructure was performed independently for each of the 7 frames of the time-resolved images acquired at the Co and Ni edges according to the algorithm described in Refs. ^{7,47}. Prior to the magnetic reconstruction, each projection was upsampled with a 2x factor using a bicubic interpolation. With this, a 3D magnetic reconstruction with a 20 x 20 x 20 nm³ voxel size was obtained, analogous to the procedure demonstrated in Refs. ⁴⁸⁻⁵⁰.

C1.11: 5) *The nature of the “destructive interference pattern” between SWs should be explained more clearly. Usually, an interference pattern indicates the presence of specific spatial points in which a constant pi phase difference results in zero oscillation amplitude (i.e. a static $\Delta\theta=0$ surface). Here, the spatial location of the $\Delta\theta=0$ surface moves in consecutive snapshots both in the experiment and the micromagnetic simulations, as corresponds to a moving wavefront rather than to an interference pattern. Also, they should explain more clearly the nature of “transient dynamics” in the context of a periodic excitation of constant amplitude. Their micromagnetic simulations only cover a fraction of the temporal period of the rf field. How is the evolution of the $\Delta\theta=0$ surface over the full excitation period?*

We thank the reviewer for these insightful comments, which allowed us to study more in-depth the character and full spatial and temporal dynamics of the interference pattern. We integrated Supplementary Information section 6: “Spatial and temporal evolution of the saddle-shaped 3D interference figure”.

To gain a comprehensive view of the phenomenon, we developed a simple general model for calculating and visualizing in three dimensions the interference pattern generated by two planar sinusoidal waves. Let us assume a reference system in which x, y are the horizontal axes (corresponding in our experiment to the CoFeB film plane) and z is the vertical axis (in our experiments the CoFeB film depth).

Consistent with the z -dependence of the amplitude of our experiments (see Fig. 3), we implemented in our MATLAB code the possibility to have a linear dependence of the amplitudes of the waves $A_1(z)$ and $A_2(z)$ along the depth z , expressed by:

$$A_1(z) = A_{1b} + \frac{A_{1t} - A_{1b}}{h} z, \quad A_2(z) = A_{2b} + \frac{A_{2t} - A_{2b}}{h} z,$$

where A_{1b} (A_{2b}), A_{1t} (A_{2t}) are the amplitudes at the surface at $z = 0$ and $z = h$, with h the total film thickness.

The full spatio-temporal profile of the two waves is therefore described by:

$$\theta_1(t, \mathbf{r}) = A_1(z) \cos(k_{x1}x + k_{y1}y - \omega_1 t), \quad \theta_2(t, \mathbf{r}) = A_2(z) \cos(k_{x2}x + k_{y2}y - \omega_2 t)$$

Where k_x and k_y are the x and y components of the wavevectors, respectively, propagating at angles α_1 and α_2 with respect to the horizontal axis, and defined as follows:

$$k_{x1} = 2\pi/\lambda_1 \cos(\alpha_1), \quad k_{y1} = 2\pi/\lambda_1 \sin(\alpha_1), \quad k_{x2} = 2\pi/\lambda_2 \cos(\alpha_2), \quad k_{y2} = 2\pi/\lambda_2 \sin(\alpha_2)$$

The frequency of the waves are:

$$\omega_1 = 2\pi f_1, \quad \omega_2 = 2\pi f_2.$$

The three-dimensional interference pattern, resulting from the superposition of the two waves is therefore:

$$\theta_{\text{int}}(t, \mathbf{r}) = \theta_1(t, \mathbf{r}) + \theta_2(t, \mathbf{r})$$

Where \mathbf{r} is the position vector. From this expression, it is possible to directly extract the amplitude of the oscillation $A_{\theta_{\text{int}}}(\mathbf{r})$ point-by-point in the three-dimensional space, via numerical Fast-Fourier-Transform. Noteworthy, in our case the wave frequencies $f_1 = f_2$, since they are fixed by the excitation field frequency, therefore $A_{\theta_{\text{int}}}(\mathbf{r})$ is constant in time. On the opposite, superposition of waves with different frequencies would give rise to time-dependent amplitudes and “beating” effects.

For reproducing the experimental conditions observed in the reconstruction of spin-wave dynamics in the CoFeB layer, we used a wavelength $\lambda_1 = \lambda_2 = 650$ nm, which is equal to the experimentally measured spin-wave wavelength, and a frequency $f_1 = f_2 = 0.86$ GHz, equal to the excitation field frequency. The thickness of the CoFeB film is $h = 40$ nm. The angles of the wavevectors were set to $\alpha_1 = 180^\circ$ and $\alpha_2 = 230^\circ$, giving rise to experimental conditions similar to Fig. 4a, where the two wavefronts are propagating from right to left at an angle of 50° with respect to each other. Regarding the wave amplitudes, we considered two z -dependent amplitude profiles $A_1(z)$ and $A_2(z)$, both decreasing linearly from the top to the bottom surface, with a slightly different decrease rate as a function of z , consistently with our findings of Fig. 3. This can originate from the different propagation angle and therefore dispersion of the two waves, their different excitation efficiency, or their interaction with the locally non-uniform texture. In this scenario, Extended Data Fig. 12 **a, b** shows top-view (xy plane) snapshot at $t = 0$ of the two individual waves.

Now let us consider the interference $\theta_{\text{int}}(t, \mathbf{r}) = \theta_1(t, \mathbf{r}) + \theta_2(t, \mathbf{r})$ generated by the spatial superposition of these two waves. Extended Data Fig. 12 **c** shows a snapshot at $t = 0$ s of the interference pattern. Since the wavefronts propagate at an angle, the resulting interference pattern features spatially alternating interference minima and maxima arising from regions with destructive and constructive interference, respectively. The direction of propagation of the interfering wavefronts is indicated as black arrows. We note that this interference figure well reproduces the experimental one shown in Fig. 4a and the one obtained via micromagnetic simulations shown in Fig. 4b. The geometry of the interference pattern is particularly clear from the corresponding “amplitude” image, which shows point-by-point the amplitude $A_{\theta_{\text{int}}}(\mathbf{r})$ of the oscillation. In particular, the central dark region marks a region where the two waves superimpose in antiphase giving rise to destructive interference and therefore 0 amplitude.

We now analyze the interference along the vertical direction z , in panels Extended Data Fig. 12 **e, f**. In panel **e**, the $\theta_{\text{int}} = 0$ surfaces are visualized in white, and feature the characteristic buried “saddle” shape we observe experimentally. It is clear that the buried saddle points (e.g. the one in green) are located where the destructive interference region intersects the perpendicular propagating wavefronts. The three-dimensional geometry of the destructive interference region is clear in the corresponding $A_{\theta_{\text{int}}}(\mathbf{r})$ image of panel **f**. Here, it is possible to appreciate the fact that the destructive interference region, where $A_{\theta_{\text{int}}} = 0$, is buried in the middle of the layer (black “tube” in panel **f**), and is composed by the points occupied by the “saddle” at different times during propagation.

Extended Data Figure 12 | Spatial and temporal evolution of three-dimensional spin-wave interference. **a, b**, Top view (xy plane) snapshot at $t = 0$ of planar wavefronts of the two individual waves. **a**, Wave 1 $\theta_1(t, \mathbf{r})$, propagating along $-x$. **b**, Wave 2 $\theta_2(t, \mathbf{r})$ propagating from right to left at an angle of 50° with respect to each other. The propagation direction is indicated by the black arrows. **c**, Top view (xy plane) snapshot at $t = 0$ s of the interfering waves $\theta_{\text{int}}(t, \mathbf{r}) = \theta_1(t, \mathbf{r}) + \theta_2(t, \mathbf{r})$. Regions of constructive and destructive interference are indicated. The arrows mark the propagation direction of the wavefronts of the interference pattern. $\theta_{\text{int}} = 0$ regions are in white. **d**, corresponding amplitude $A_{\theta_{\text{int}}}(\mathbf{r})$ map, where destructive (constructive) interference gives rise to amplitude minima (maxima). **e**, z -dependence of the interference patterns, featuring the buried saddle-shaped $\theta_{\text{int}} = 0$ surface in correspondence of the green point. As a function of time, the saddle and the wavefronts propagate in the direction of the arrows. The dashed rectangle marks a single saddle, for comparison with Fig. 4c. **f**, Corresponding z -dependent $A_{\theta_{\text{int}}}(\mathbf{r})$ amplitude profile, displaying a “buried” destructive interference region (black “tube”) located in the middle of the layer, characterized by $A_{\theta_{\text{int}}} = 0$.

We clarified these concepts in the main manuscript and refer to the above new supplementary section:

To gain further insight on the dynamics within the interference, we performed micromagnetic simulations, shown in Fig. 4e, f (see also Extended Data Fig. 10-11 and Movie S10), and developed a general model for calculating and visualizing in three dimensions the interference

pattern generated by two planar sinusoidal waves with z -dependent amplitudes, which nicely reproduce the three-dimensional pattern we observe experimentally. We find that the saddle-shaped $\Delta\theta = 0$ surface in CoFeB is a propagating dynamic structure, located in correspondence of a “buried” destructive interference region, where the spin-wave amplitude is zero (more details in Extended Data Fig. 12 and related discussion). This originates from the interference of spin waves with different depth-dependent dynamics, giving rise to a distribution of minima and maxima within the volume, and to three-dimensional features within the propagating wavefronts, such as our saddle-shape structure.

In summary, I consider that this work reports on a very interesting advance of vector imaging of spin waves with good spatial and temporal resolution, revealing the complex 3D configuration of magnons in magnetic multilayers that had not been reported before. In my opinion, this work will deserve publication in Nature Communications, once the points described above are properly clarified.

We thank the reviewer for appreciating our work.

Reviewer #2:

Dear editor and authors, please find in the following my review of the current state of the manuscript, starting with a short description of its content and findings:

In the presented manuscript, the authors showcase the magnetization imaging technique 'Time-Resolved Soft X-ray Laminography' on spin-wave dynamics in microstructures of a synthetic antiferromagnet.

For adequate samples, mainly thin enough to be sufficiently x-ray transparent, a rotation about an oblique axis (here taken for 37 angles at 45 degree tilt) allows capturing the magnetization with claimed nanoscale spatial resolution and sub-ns (70ps,160ps) temporal resolution. Furthermore, selecting photon-energies matching a materials L2 edge allows element specific signal collection.

The technique has been demonstrated previously in [1], [2] and utilized on magnetization dynamics of a domain wall and vortex gyration [3]. It bears great similarities to soft x-ray tomography techniques able to resolve the full magnetization vector field.

Despite its previous principled demonstration on magnetization dynamics it is in my opinion an indeed interesting and important progression to apply this imaging technique to other related magnetization dynamics, such as that for spin waves, in this case on synthetic antiferromagnets.

In particular as the spin-wave dynamics are substantially modified by coupling between layers and their local amplitude distribution should give insights into local interfacial torques, inhomogeneities, defects etc.

The work addresses these dynamics in rectangular microstructures of the synthetic antiferromagnet CoFeB(50nm)/Ru(0.5nm)/Ni(40nm). Large changes in the ellipticity of precession across the film thickness, and of the precession amplitude in particular near the common spacer-interface are recorded.

*This is also expected based on the current understanding and micromagnetic simulations. A good agreement with these micromagnetic simulation and experimental results is observed throughout the addressed regions of the sample. The data is clearly presented and appears of high quality. **Comment 2.1:** In my opinion the manuscript would greatly benefit from addressing the differences from the idealized micromagnetic simulation in particular on the -x,-y quadrant, where the expected landau pattern is heavily distorted. The authors suggest variations of the interlayer exchange coupling to be the reason. Such differences should also change the spin-wave mode profile considerably, allowing to characterize these differences further based on the experimental data in tandem with micromagnetic simulations of different coupling constants. However, I do not view inclusion of this as a necessity for publication.*

In the current state, the manuscript successfully demonstrates a high degree of agreement of the recorded magnetization dynamics with theoretical expectations and micromagnetic simulation such as general propagation and complex interference patterns.

This confirms the capability of the measurement technique to capture all dynamics accurately and the match with the simulation corroborates this as it is based on the previous demonstrations of the technique and other studies on spin waves in synthetic antiferromagnets.

In this respect, I recommend the work for publication with a few minor points to address:

We thank the reviewer for their appreciation of our work and its quality and for their review which helped us to improve our manuscript. In addition, especially to address the referees' questions on the achievable spatial resolution of the technique, which is a multi-faceted topic, we have performed additional analysis of the acquired data. As detailed in our answer to comment 1.10, we performed an upsampling of the acquired experimental data to obtain a 3D reconstruction with a voxel size closer to the theoretical limit of the achievable spatial resolution. The results obtained via 3D imaging are confirmed by micromagnetic simulations.

In the following we address the minor points raised by the reviewer.

Description of the micromagnetic simulations:

C2.2: *I would recommend to clarify the micromagnetic simulations for the different regions in particular region 4. The type of excitation and free propagation path prior to the observed spin-wave interference would be illuminating to the readers.*

We thank the reviewer for this suggestion, below we integrated our description of the micromagnetic simulations reported in the Methods section with some details about the excitation.

In particular, we added:

Spin waves were excited by using narrow line excitations, where a sinusoidal out-of-plane magnetic field oscillating at a frequency of 0.86 GHz was applied. The magnitude of the excitation field was in the 5 mT – 7 mT range for all the simulations. In order to simulate the z-dependence of the SW amplitudes (Figure 3d, e), a rectangular stripe geometry of $20.48 \mu\text{m} \times 20 \text{nm} \times 90 \text{nm}$ was employed. The total simulated volume was discretized into cells having dimensions of $5 \times 5 \times 5 \text{nm}^3$. In this case, spin waves were excited by using a single narrow line excitation in the middle of the rectangular stripe. For simulating the 3D interference (Figure 4b, e, f), a $2.5 \mu\text{m} \times 2.5 \mu\text{m} \times 90 \text{nm}$ volume with cell dimensions of $5 \times 5 \times 5 \text{nm}^3$ was employed. Spin waves were excited by using two single narrow line excitations positioned at a 50° angle between each other. Periodic boundary conditions within the plane were used in both cases.

Regarding the free propagation path, we simulated a length of about 1200 nm from emitters (about two wavelengths, consistently with the experimental data). The direction of propagation, i.e. the spin-wave wavevector, results perpendicular to the excitation line for this acoustic modes. During the propagation, the wavefronts are planar, as in the experiments, as can be seen from the top view of the image in Fig. 4b which has been reported for clarity below:

b Micromagnetic simulations - CoFeB

To gain a comprehensive view of the interference phenomenon, we also developed a simple general model for calculating and visualizing in three dimensions the interference pattern generated by two planar sinusoidal waves, which accurately reproduces our experimental findings. Please, see comment C1.11 and the new Extended Data Fig. 12.

C2.3: *SI Figure 8: the color bar appears saturated making it impossible to read out the profile in the saturated regions from the given images. As the saturated regions take up more than 90% of the image, it would be certainly beneficial to find a solution. Either via a second separate color bar or logarithmic plotting etc.*

In Extended Data Fig. 10, we used a saturated color bar in order to unequivocally identify with precision the geometry and position of the $\Delta\theta = 0$ region as we approach the saddle region. For comparison, we show here and add to the revised version of the Supplementary Information Extended Data Fig. 11 with three vertical sections of the saddle-shaped interference, with less saturated color scale. In addition, in order to fully visualize the interference figure in 3D, we developed a general model for calculating the interference region. In particular, in Extended Data Fig. 12, we show the full three-dimensional interference in correspondence of the saddle region, where the lobe shape and sinusoidal profile can be better appreciated.

Extended Data Figure 11 | Spatial dependence of the saddle-shaped three-dimensional interference pattern (soft color scale). **a**, Snapshot of time-resolved micromagnetic simulations of a saddle-shaped interference region. Scale bar, 200 nm. **b**, Vertical sections extracted in correspondence of the black lines in panel **a**. The color code indicates the in-plane dynamic angle $\Delta\theta$. Scale bars, 45 nm.

C2.4: *SI Figure 6: The caption should be rephrased in my opinion as it currently reads a bit confusingly, but is technically fine. In b) the title should read $A(\Delta\theta)$ if I understand the caption correctly.*

The caption has been updated to enhance its clarity. Below the revised version:

Extended Data Figure 6 | Simulations of the SW amplitude and localization through the film thickness in a compensated SAF. **a**, Sketch of the simulated compensated SAF structure comprising two 45-nm-thick CoFeB layers, antiferromagnetically coupled. **b,c**, Relative variations calculated with respect to the bottom surface of the in-plane SW amplitude $A_{\Delta\theta}$ (**b**) and out-of-plane SW amplitude $A_{\Delta\phi}$ (**c**) as a function of the z-position through the thickness of the sample. The simulated SW frequency was $f = 0.86$ GHz.

C2.5: *SI Figure 5: In my opinion this figure nicely clarifies, the extracted line profiles and could be moved up as additional information added to figure 1. Alternatively, the extraction region e.g. of figure 2 could/should be visualized there for clarification in my opinion.*

We thank the reviewer for the suggestion. We have updated the figure accordingly:

Figure 1 | Experimental setup and time-resolved three-dimensional reconstruction of spin waves in a synthetic antiferromagnet. [...] **f**, Experimental reconstruction of the three-dimensional magnetization dynamics (arrows) excited at 0.86 GHz in NiFe and CoFeB. In this snapshot, spin waves are visualized as sinusoidal spatial oscillations of the in-plane magnetization dynamics ($\Delta\theta$, blue-red coloring). **The regions of the sample studied in each figure are indicated in yellow.** On the right, planar spin waves emitted by a domain wall (dashed line) propagate along k_{SW} . On the left, spin waves propagating at an angle interfere in the dashed rectangular region corresponding to Fig. 4. Scale bar, 200 nm.

As mentioned above, I believe the manuscript would greatly benefit from studying the distorted regions of the structure in regard of the information that can be gained by analyzing the local spin-wave precessions to infer on the type of defect/coupling change/ etc in those regions. However, as mentioned I do not view this as a necessity, rather a suggestion and believe that such imaging studies are of high value to the field of magnonics and its development into 3D structures.

We thank the reviewer for their suggestion, indeed it would be very interesting to study this additional non-ideal regions to highlight the 3D properties on the spin waves there. We plan to address this topic in a future work.

[1] C. Donnelly *et al.*, “Time-resolved imaging of three-dimensional nanoscale magnetization dynamics,” *Nat. Nanotechnol.*, vol. 15, no. 5, pp. 356–360, May 2020, doi: 10.1038/s41565-020-0649-x.

[2] K. Witte *et al.*, “From 2D STXM to 3D Imaging: Soft X-ray Laminography of Thin Specimens,” *Nano Lett.*, vol. 20, no. 2, pp. 1305–1314, Feb. 2020, doi: 10.1021/acs.nanolett.9b04782.

[3] S. Finizio, C. Donnelly, S. Mayr, A. Hrabec, and J. Raabe, “Three-Dimensional Vortex Gyration Dynamics Unraveled by Time-Resolved Soft X-ray Laminography with Freely Selectable Excitation Frequencies,” *Nano Lett.*, vol. 22, no. 5, pp. 1971–1977, Mar. 2022, doi: 10.1021/acs.nanolett.1c04662.

References

1. Grünberg, P. Magnetostatic spin-wave modes of a heterogeneous ferromagnetic double layer. *Journal of Applied Physics* **52**, 6824–6829 (1981).
2. Wintz, S. *et al.* Magnetic vortex cores as tunable spin-wave emitters. *Nature Nanotech* **11**, 948–953 (2016).
3. Sluka, V. *et al.* Emission and propagation of 1D and 2D spin waves with nanoscale wavelengths in anisotropic spin textures. *Nat. Nanotechnol.* **14**, 328–333 (2019).
4. Albisetti, E. *et al.* Optically Inspired Nanomagnonics with Nonreciprocal Spin Waves in Synthetic Antiferromagnets. *Adv. Mater.* **32**, 1906439 (2020).

Reviewers' Comments:

Reviewer #1:

Remarks to the Author:

Girardi et al have properly revised their manuscript along the lines indicated in my previous report, providing additional information on their measurement and data analysis procedures, and including additional experimental information regarding individual SW time dependence and direct microscopy images. In addition, they have developed a model that gives a very good insight into the novel physics of SW interference in 3D.

Therefore, I recommend the publication of this work in Nature Communications in its present form.

Reviewer #2:

Remarks to the Author:

I thank the editor and authors for their work and find my questions/points have been answered to satisfaction and clarifications are included. Hence, i recommend the manuscript for publication. Kind regards